# Islet Regeneration: Endogenous and Exogenous Approaches

**DOI:** 10.3390/ijms22073306

**Published:** 2021-03-24

**Authors:** Fiona M. Docherty, Lori Sussel

**Affiliations:** Barbara Davis Center for Diabetes, University of Colorado Anschutz Medical Campus, Aurora, CO 80045, USA; Fiona.Docherty@cuanschutz.edu

**Keywords:** diabetes, pancreas, islet, beta cell, regeneration

## Abstract

Both type 1 and type 2 diabetes are characterized by a progressive loss of beta cell mass that contributes to impaired glucose homeostasis. Although an optimal treatment option would be to simply replace the lost cells, it is now well established that unlike many other organs, the adult pancreas has limited regenerative potential. For this reason, significant research efforts are focusing on methods to induce beta cell proliferation (replication of existing beta cells), promote beta cell formation from alternative endogenous cell sources (neogenesis), and/or generate beta cells from pluripotent stem cells. In this article, we will review (i) endogenous mechanisms of beta cell regeneration during steady state, stress and disease; (ii) efforts to stimulate endogenous regeneration and transdifferentiation; and (iii) exogenous methods of beta cell generation and transplantation.

## 1. Introduction

According to the latest figures from the International Diabetes Federation (IDF), approximately 463 million adults are living with diabetes, and this number is estimated to rise to 700 million individuals by 2045 (https://www.idf.org/aboutdiabetes/what-is-diabetes/facts-figures.html; accessed on 19 March 2021). The majority of these cases represent type 2 diabetes (T2D); whereas type 1 diabetes (T1D) constitutes approximately 5–10% of all cases of diabetes. In both T1D and T2D, there is a marked reduction in functional beta cell mass. In individuals with T1D, the primary cause of beta cell loss is through the autoimmune destruction of beta cells. In T2D patients, both a reduction in beta cell mass and impaired beta cell insulin secretory function have been observed [1,2,3]; however, whether the decrease in beta cell mass contributes to beta cell dysfunction or the dysfunction is secondary to decreased mass, is not clearly understood. Since T2D represents a heterogeneous disease, it is likely that either scenario can contribute to disease, depending on the individual. Regardless of the primary defect, stabilizing functional beta cell mass in both T1D and T2D patients will be key to developing a therapy for these increasingly prevalent diseases. Viable treatment options could involve preserving beta cells, replacing beta cells from endogenous sources and/or replacing lost beta cells with an exogenous source of stem cell-derived beta cells. To be successful in any of these endeavors, however, it will be necessary to first understand the normal mechanisms of beta cell loss and regeneration in the developing and adult pancreas.

## 2. Beta Cell Mass Reduction in Diabetes

Beta cell mass predominantly depends on the combined rates of beta cell replication, neogenesis and hypertrophy minus the rate of beta cell apoptosis. Beta cell mass is generally quantified by the analysis of histology samples—measuring beta cell area and normalizing to total pancreas area (for beta cell volume) or normalizing to total pancreatic weight (for beta cell mass)—confusingly, beta cell mass and volume tend to be used interchangeably in the literature [4]. In T1D, immune-mediated apoptosis is the primary cause of beta cell loss; however, a recent study showed that both beta cell mass and exocrine cell numbers were already decreased in individuals with T1D prior to or shortly after disease onset [5]. In T2D, it has been suggested that a decrease in beta cell mass is primarily caused by an increase in beta cell apoptosis that outweighs beta cell proliferation [6], resulting in inadequate insulin production. T2D patients have approximately 30% lower beta cell mass compared to non-diabetic individuals of the same BMI [7]. With as little as a 1.1% reduction in beta cell mass, fasting blood glucose levels become elevated [3]; a phenomenon observed in both lean and obese patients with T2D [8]. Importantly, it is not necessarily the absolute amount of beta cell mass that is important, but whether there is sufficient beta cell mass to appropriately respond to metabolic demand [9].

Although a reduction in beta cell mass has predominantly been attributed to beta cell death, studies in mouse models of both T1D and T2D have suggested that beta cell dedifferentiation and/or transdifferentiation can also contribute to beta cell loss [10,11,12]. There is some evidence of similar mechanisms occurring in islets of diabetic patients [13,14]; reviewed in [15]; however this is more difficult to demonstrate in the absence of in vivo lineage tracing. If indeed a component of beta cell loss is due to loss or alterations in beta cell identity, mechanisms to restore beta cell identity could represent alternative therapeutic options.

With the increasing focus on beta cell dysfunction and loss as a contributor to the progression of both T1D and T2D, extensive efforts have been made to identify mechanisms to replace functional beta cell mass. As discussed below, many studies have explored how beta cell populations are established and expanded during different critical windows of life, including fetal development, adult basal function, as well as injury and stress conditions, with the goal of developing therapeutics that can promote the induction of these mechanisms to restore beta cell mass in afflicted individuals. These studies have met with some success and, although several controversies remain, they have revealed important insights into the biology of rodent and human beta cell development and growth.

## 3. Intrinsic Beta Cell Mass Expansion

### 3.1. Establishment of Beta Cell Mass

The establishment of beta cell mass in rodents and humans occurs predominantly during embryonic and early postnatal stages [16,17,18]. In rodents, where beta cell mass has been studied in depth due to the ready availability of histological samples, beta cell mass is established through a combination of neogenesis and proliferation during development (Figure 1). In mice, beta cells are initially derived from the Neurogenin3 (NEUROG3) expressing endocrine progenitor population, followed by the perinatal expansion of differentiated beta cells, primarily through proliferation [19,20,21]. The relative contributions of neogenesis and beta cell replication in rat (in which beta cell mass has been shown to increase by two-fold in the first five days after birth) have also been studied using bromodeoxyuridine (BrdU) as a lineage tracing tool. BrdU was introduced into beta cells at postnatal day two, followed by the quantification of BrdU positive cells present at day five. This study demonstrated that 12% of the beta cells present at day five were derived from existing beta cells [22]. Several additional studies have confirmed that a large proportion of differentiated beta cells are established by birth, followed by a burst of beta cell growth in neonates—predominantly due to the proliferation of existing beta cells, with a smaller contribution from neogenesis (Figure 1; reviewed in [23,24]. In humans, beta cell differentiation is first observed at gestational week nine and increases linearly until birth, with evidence suggesting that beta cell expansion depends on both neogenesis and proliferation [25]. In basal conditions, the postnatal expansion of the human beta cell population continues, largely through proliferation, and peaks around two years of age [18]. The same study suggested that baseline beta cell mass was established by five years of age, based on the analysis of human pancreatic tissue derived from 40 individuals spanning from fetal tissue to 70 years of age.

There have been many studies, primarily in rodents, to identify endogenous extrinsic regulators of beta cell mass during the highly proliferative neonatal period, with hopes that these factors could be harnessed to promote beta cell proliferation on demand. In particular, factors supplied from the islet extracellular microenvironment have been shown to be necessary for beta cell proliferation. Some of the earliest studies focused on components produced by the extracellular matrix (ECM) and vascular structures surrounding the islets. In vitro studies identified endothelial cells as a source of laminins that promoted proliferation of the immortalized mouse insulinoma 6 (MIN6) cell line and suggested beta1-integrin was required for this response [26]. Subsequent studies demonstrated that beta1-integrin is produced at high levels in the perinatal pancreas within both the ECM and vascular structures surrounding islets and in lower levels within the beta cells themselves, suggesting a role for cell–cell and cell–matrix signaling [27]. This hypothesis was supported by in vivo mouse studies that showed ablation of beta1-integrin specifically within beta cells disrupted their adhesion to ECM substrates and resulted in an 80% reduction in beta cell mass [27]. These mice displayed impaired perinatal islet proliferation, without significant changes in islet function.

Neonatal pancreatic pericytes, which are components of the islet vasculature, have also been shown to be necessary for neonatal beta cell proliferation. Using an in vivo diptheria toxin ablation strategy to delete pericytes in neonatal mice, Ephstein et al. showed the depletion of islet-associated pericytes significantly reduced beta cell proliferation [28]. Furthermore, ex vivo reconstitution experiments with both pericytes and pericyte conditioned media, demonstrated that pericytes were able to stimulate beta cell proliferation. These studies also implicated vascular basement membrane components that are produced by pericytes as important beta cell pro-proliferative molecules [29]. Many additional studies have gone on to explore the role of ECM and vasculature components in promoting proliferation in the neonatal islet and are reviewed extensively in [30,31].

Islet associated macrophages have also been implicated in promoting neonatal beta cell proliferation. Mussar et al., used in vivo depletion and reconstitution experiments in mice to demonstrate that a CCR2-positive subpopulation of macrophages were transiently enriched in the neonatal pancreas and produced the growth factor IGF2 to promote proliferation [32]. The depletion of this population caused impaired beta cell proliferation, in addition to islet dysfunction and glucose intolerance; beta cell mass was restored when these cells were reintroduced into mice through adoptive transfer strategies. In adult mice, VEGF-A-mediated recruitment of macrophages could also induce limited beta cell proliferation [33]. Although these studies to identify islet microenvironment factors that promote proliferation of neonatal beta cells have been fruitful, their influence on adult beta cell proliferation has been less effective, suggesting there are intrinsic beta cell properties that are lost or gained upon maturation that make them more resistant to proliferation in response to extrinsic signals (see the mechanism of beta cell replication, below).

Once formed, beta cells are extremely long-lived with limited evidence of expansion in basal conditions [16,34]. However, there is evidence for a small amount of turnover in the adult islet that has been postulated to result from both beta cell proliferation and neogenesis, although the relative contribution of the respective mechanisms has been widely debated. Several studies have provided convincing evidence for beta cell proliferation using thymidine labeling [17,34,35,36]. The use of an elegant Cre-lox lineage tracing system in which beta cells were heritably labelled with alkaline phosphatase and observed over time for the appearance of non-stained beta cells also suggested that in rodents, adult beta cell mass is largely maintained through the replication of preexisting beta cells [35]. It should be noted, however, that the basal rate of replication has been shown to decline with age; with fewer than 0.1% of beta cells undergoing replication [37], which could suggest the existence of an alternative mechanism for the retention of beta cell mass in older animals.

The case for neogenesis, especially in basal conditions, has been much more controversial, with almost an equal number of studies providing evidence to support the neogenesis hypothesis as there are refuting it (reviewed in [38]). One major hindrance to these studies has been the failure to identify a stem cell marker that would allow for lineage tracing to clearly identify newly formed beta cells. With the advent of single cell technologies, however, evidence of rare populations of progenitor-like cells has emerged. For example, van der Meulen et al. have identified a population of immature beta cells within a specialized micro-environment at the islet periphery [39]. These cells appear to be highly plastic in their identity with the potential to contribute to both mature alpha and beta cell populations throughout life. Ultimately, since beta cell turnover in basal conditions is a relatively rare event, it is likely that there is a small contribution from both mechanisms to maintain beta cell mass throughout life. As discussed below, stress or injury conditions may be necessary to reveal the full regenerative potential of beta cells.

### 3.2. Postnatal Expansion of Beta Cell Mass in Humans

Although there has been an increase in access to human cadaveric tissue in recent years, tools to investigate beta cell mass in humans remains extremely limited. Live imaging is very challenging due to the position of the pancreas in the abdomen, the small size of islets, and the lack of specific beta cell markers. Significant research efforts have explored the use of non-invasive imaging modalities, including positron emission tomography (PET), computed tomography (CT), and magnetic resonance imaging (MRI), and although these methods have been moderately successful for monitoring transplanted islet mass, they have been less efficacious for measuring endogenous beta cell mass (reviewed extensively in [40,41]). Pancreatic biopsies are not a viable option due to the dangers associated with collecting the samples; as such, post-mortem pancreas samples are the primary source for studying the dynamics of beta cell mass in humans. Studies using a combination of abdominal CT scans of pancreata from individuals between the ages of four weeks and 20 years, combined with the morphometric analysis of autopsy pancreatic tissue obtained from children age two weeks to 21 years, suggested that beta cell mass increased by several fold between birth and adulthood in humans [25]. Notably, it appeared that this expansion was associated with increases in islet size rather than islet number. The rate of beta cell replication is highest in early childhood, after which it becomes almost undetectable. Consistent with this time course, immunocytochemistry analysis of cell cycle proteins has revealed that the peak of beta cell expansion occurs at around two years of age; at this timepoint, 2.6% of total beta cells are engaged in the cell cycle [18,25]. Further studies have suggested that replication is mediated by age-dependent cell cycle regulation in beta cells with the expression of the key factors p16 and p27 being down regulated in beta cells with aging. The occurrence of neogenesis in human islets has also been speculated due to the close association of small islets with the pancreatic duct [42,43]; however, in the absence of endogenous lineage tracing, the existence of islet progenitor cells in the adult have been more difficult to identify. A recent study using a 3D culture system using islet depleted human pancreatic tissue identified a progenitor population that expressed Lgr5 and exhibited high aldehyde dehydrogenase activity that could be differentiated into insulin-producing cells in vitro and upon transplantation [44]. Additional studies will be needed to determine whether this population has similar regenerative potential in vivo. 

## 4. Metabolic Stress Conditions Promote Increase in Beta Cell Mass

Although regeneration in the adult pancreas in basal conditions is limited, it has been shown that in conditions associated with metabolic stress, such as pregnancy and/or obesity, the endocrine compartment can compensate by expanding beta cell mass, including through the increase in beta cell size (hypertrophy), altering the balance between beta cell proliferation and apoptosis, and/or producing new beta cells through neogenesis or transdifferentiation (Figure 1). These mechanisms have been predominantly explored in rodent models, however, there is accumulating data on the occurrence of similar mechanisms in humans.

### 4.1. Beta Cell Mass in Metabolic Stress—Pregnancy

Reversible beta cell mass expansion occurs during pregnancy to compensate for progressive insulin resistance; failure to compensate results in gestational diabetes. Interestingly, 70% of women who develop diabetes while pregnant will go on to become diabetic later in life, emphasizing the importance of adaptive changes in beta cell mass for preventing both gestational and T2D [45]. In rodents, it has been shown that a ~three to four-fold increase in beta cell mass is due to both beta cell hypertrophy and beta cell proliferation [46,47,48]. Perhaps the most intriguing part of the beta cell expansion during the pregnancy paradigm is that it is completely reversible; in rodents, beta cell mass returns to normal levels within ten days after birth. This rapid decrease in beta cell mass is achieved by increased apoptosis, decreased proliferation, and the hypotrophy of existing beta cells [49].

Many studies have explored the mechanisms underlying beta cell expansion in the rodent pregnancy model, aiming to identify and harness signaling factors that induce beta cell mass expansion that could represent a powerful mechanism to promote beta cell expansion to treat beta cell loss in diabetes. Some of the earliest studies correlated beta cell proliferation with the onset of placental lactogen and/or prolactin secretion [47,50]. Additional studies have demonstrated a causal role for placental lactogens and prolactin promoting both beta cell proliferation and insulin secretion [50,51,52,53]; as well as demonstrating through genetic inhibition that the prolactin receptor is essential for the observed adaptive responses [54,55,56]. Downstream effectors of placental lactogens and prolactin have also been identified, including serotonin and the transcriptional regulator FoxM1 [57]; however, applying this information to promote the expansion of human beta cells has met with limited success, likely because of innate differences in rodent and human physiology.

### 4.2. Beta Cell Mass in Metabolic Stress—Obesity

Obesity is associated with peripheral insulin resistance that requires increased insulin production to prevent the development of hyperglycemia. This increase is primarily achieved by a compensatory increase in beta cell mass and insulin output that allows the maintenance of glucose homeostasis; T2D ensues when this compensation fails [6,8,58,59,60]. In rodent models, obese nondiabetic mice display a compensatory increase in beta cell mass [61] and it is believed that beta cell proliferation plays a key role in this expansion of beta cell mass. Evidence for proliferation was shown in several diet induced or genetic models of obesity that are able to compensate for increased metabolic demand by increasing beta cell proliferation [62,63,64]. Although there is compelling evidence that compensatory beta cell expansion occurs—at least early in the insulin resistant state—the systemic circulating factors that trigger this expansion have been difficult to identify. Many groups have focused on general growth factors, including members of the fibroblast growth factor (FGF) and hepatic growth factor (HGF) families and insulin and insulin-like growth factors (IGF) (reviewed in [65]); however, the in vivo application of these factors has proved challenging due to their global mitogenic activities. The Kulkarni group took a more directed approach by using the liver-specific insulin receptor knockout (LIRKO) mouse as a model of insulin resistance to characterize cross-talk between the liver and pancreatic islets [66]. These studies demonstrated that circulating factors derived from the liver could stimulate the proliferation of mouse and human islets. Among these, they identified SerpinB1, a highly conserved liver-derived protease inhibitor that could stimulate beta cell proliferation in zebrafish, mice and humans through the activation of the insulin/IGF signaling cascade. In addition to signals from the liver, several in vitro and in vivo studies have demonstrated that the gut secreted protein, Glucagon-like peptide 1 (GLP-1) and its analogs can stimulate beta cell proliferation in rodent models and in humans. Many studies are now ongoing to determine the downstream intracellular pathways that are activated in response to GLP-1 to more effectively promote increased beta cell mass as a treatment for diabetes [67].

One challenge in the field has been to understand the intracellular pathways that facilitate or—more often—impede beta cell proliferation. In one approach, transcriptional analysis demonstrated the differential regulation of cell cycle genes required for cell proliferation between the C57BL/6 mice that can increase their beta cell mass and BTBR mice—a strain that is unable to increase beta cell mass sufficiently to meet the insulin demand and as a result develops diabetes [64]. Although less data are available in the human context, there is evidence from cadaveric tissue that increased beta cell mass also occurs [68]; however, it is more difficult to determine whether beta cell mass expansion in obesity is primarily due to proliferation. Studies from the Butler lab were unable to detect differences in beta cell proliferation between lean, obese and T2D individuals using Ki67 staining of cadaveric pancreatic tissue [6,60]. However, there are conflicting reports in studies using proliferating cell nuclear antigen (PCNA) staining [58,69]. Davis et al. [70] have also shown that numerous cell cycle genes, including Ki67, are upregulated in islets of obese individuals; whereas other studies suggest that these proteins are cytoplasmic rather than nuclear localized in beta cells [71,72]. Again, the lack of non-invasive modalities to measure beta cell mass dynamics in humans prevents the feasibility of performing longitudinal studies that could resolve some of these controversies.

## 5. Models of Beta Cell Mass Expansion

In addition to animal models of long-term stress conditions, such as pregnancy or insulin resistance, artificial models of beta cell mass expansion involving experimentally induced injury have become powerful tools to study tissue remodeling in the pancreas. These studies have been instrumental in characterizing the response to acute stress and have revealed previously unappreciated cellular plasticity in both the rodent and human pancreas. 

### 5.1. Partial Duct Ligation: The Case for Neogenesis?

Partial duct ligation (PDL) is a model of severe pancreatic injury that has been used in rodents for many decades to investigate the plasticity of the pancreas. The model is generated by ligating one of the main ducts responsible for draining enzymes secreted by acinar cells to cause the inflammation and atrophy of approximately 50% of the acinar cells proximal to the ligature. Several studies have shown that PDL also results in an expansion of beta cell mass [73,74], though controversy exists around how these new beta cells arise. Xu et al. were among the first to suggest that in response to PDL, the re-activation of the embryonic endocrine progenitor marker NEUROG3 in ductal cells contributed to the generation of new beta cells that repopulated the islet. However, several subsequent studies suggested that PDL could not induce duct-derived beta cells in the adult islet [75,76]. The controversy was compounded when Van de Casteele et al. used lineage tracing to demonstrate beta cells could differentiate from a PDL-activated NEUROG3^+^ progenitor, although only a subset of cells re-activated *Neurog3* and to a lesser extent than previously described [77]. However, an alternative lineage tracing system suggested that PDL was only able to induce NEUROG3 expression in pre-existing beta cells rather than ducts, and there was little evidence that ductal NEUROG3^+^ cells were able to contribute to neogenesis [78]. To date, the debate continues—most recently with the determination of extensive heterogeneity within pancreatic cells, including the ductal populations [79,80,81].

### 5.2. Partial Pancreatectomy: Dedifferentiation/Redifferentiation?

Experiments involving partial pancreatectomy (Ppx) in rats showed that in conditions of chronic hyperglycemia, the remaining beta cells downregulate the expression of key beta cell associated genes such as insulin, GLUT2 and glucokinase and upregulate the expression of so-called “disallowed genes” (i.e., those associated with a non-beta cell fate) in response to this acute stress [82,83,84]. These cells were also shown to have severely impaired insulin secretion in response to glucose stimulation, which was interpreted to be caused by the dedifferentiation of beta cells to a less mature state. Subsequent studies determined that many of these disallowed genes were also upregulated in the islets of T2D individuals. Although the extent that dedifferentiation contributes to loss of beta cell mass has been more difficult to verify in humans, several single cell RNA-seq studies on T2D islets and the generation of genetic mutations in mice have identified a number of transcriptional regulators that could contribute to the loss of beta cell functional maturity, including PDX1, FOXO1, and NEUROD1 [11,14,85,86,87,88]. Although additional studies will be necessary to understand the molecular mechanisms associated with the dedifferentiation process, the realization that beta cell dedifferentiation could contribute to the number of beta cells with decreased functionality in T2D suggests it may be possible to develop alternative regenerative therapies focused on the “re-differentiation” of the beta cell population.

### 5.3. Beta Cell Ablation Models: Plasticity?

One of the more surprising observations in recent years—first in genetic mouse models and now with some validation in human tissues—is the remarkable plasticity of adult endocrine cells. Several mouse genetic models in which pancreatic transcriptional regulators have been deleted have revealed robust transdifferentiation between the alpha and beta cell lineages [89,90,91,92]. There are also now several examples in which islet cells can undergo intercellular conversion upon exposure to stress. In rodent models of severe diabetes (i.e., beta cell ablation), alpha cells became transdifferentiated into functional beta cells [93]; although their incapacity for proliferation compromised their ability to completely rescue the diabetic state in terms of beta cell number. Interestingly, alpha cells were only able to transdifferentiate early in life, while delta cells were capable of converting to beta cells later in life [94]. This suggests there may be a temporal window for certain intra-islet plasticity in the rodent system. van der Meulen et al. have also proposed the intriguing idea that the immature beta cell population located within a “neogenic niche” may represent an intermediate stage in the transdifferentation of alpha cells to beta cells [39]. Although most examples of interconversion between the islet lineages come from rodent models, there is evidence of beta cells co-expressing insulin and other endocrine hormones in T2D islets, which may represent a transient stage in the transdifferentiation process [95,96]. Again, evidence of islet plasticity raises the exciting possibility of alternative regenerative strategies for restoring functional beta cells in diabetes.

## 6. Mechanisms of Beta Cell Proliferation

Although beta cell replication is a rare event, evidence that it can be stimulated in certain conditions raises the hope that the augmentation of beta cell proliferation could be a potential regenerative therapy. Studies of natural and artificial models of beta cell mass expansion in rodents have begun to shed light on the molecular mechanisms involved, providing essential knowledge required to translate these findings to utilize proliferation as a therapeutic that promotes human beta cell proliferation.

### 6.1. Cell Cycle Regulators

One obvious area of interest is cell cycle regulators and how they can promote or suppress beta cell replication. In rodents, Georgia and Bhushan demonstrated that Cyclin D2 was uniquely required for beta cell replication in the early postnatal period. It was expressed at high levels in beta cells up to two weeks postnatally but was downregulated in adult beta cells [17]. Furthermore, the deletion of Cyclin D2 resulted in the impaired postnatal expansion of beta cells. A subsequent study by Kushner et al. demonstrated that combined functions of CyclinD1 and D2 were necessary for postnatal beta cell proliferation [97]. This might suggest that reactivating these cyclins would be sufficient to promote beta cell replication in adults; and indeed there are reports that the induced expression of Cdk4, which has been shown to partner with Cyclin D2 [98], was able to promote increased beta cell mass [99,100]. However, even in these rodent models, the expansion of beta cell mass is modest; likely due to the presence of cell cycle inhibitors, such as p16INK4a that functions to constrain uncontrolled replication [101,102,103]. Furthermore, it has been shown that the induction of beta cell replication results in reduced functionality, limiting the value of forced beta cell expansion [104]. Interestingly, the identification of heterogeneous beta cell populations has also demonstrated a negative correlation of proliferative ability and function in vivo; replication-competent beta cells tended to be less functional than their replication-incompetent counterparts [105]. The existence of these populations suggests that the ability to influence the interconversion between these beta cell populations would allow for therapies that first promote beta cell replication to expand a subset of less functional beta cells, followed by a means to return them to a less proliferative, but more highly functional state.

Several studies have also explored the differential expression of cell cycle regulators in rodent versus human islets to understand why rodent beta cells have increased replicative potential. A comprehensive evaluation of cell cycle regulators in human islets revealed that although cell cycle regulators required for entrance into cell cycle are expressed in beta cells, they are sequestered in the cytoplasm [72]. Additional studies have implicated epigenetic mechanisms in creating a global genetic environment that inhibits beta cell replication [106,107].

### 6.2. Transcriptional Regulation of the Cell Cycle

In efforts to understand and perhaps harness the upstream regulators of the cell cycle machinery, several groups have explored the role of transcription factors. One of the most promising regulators of cell cycle initiation and progression in mouse and human beta cells is the forkhead box transcription factor FOXM1. Several studies have demonstrated the importance of FOXM1 for beta cell proliferation in rodents following injury or in pregnancy [57,108]. Furthermore, FOXM1 expression positively correlated with human body mass index and was able to stimulate human beta cell proliferation by activating the cell cycle [70]. The transcriptional regulator MYC has also emerged as an important regulator of rodent beta cell proliferation by directly regulating cyclin expression in response to increased metabolic demand [109,110]; however, the upregulation of MYC was not sufficient to induce beta cell replication in older mice fed a high fat diet [111]. Although high levels of MYC expression are often associated with tumorigenesis, moderate MYC induction in human islets has also been correlated with increased beta cell proliferation [109].

NKX6.1, an essential pancreatic homeobox transcription factor, is highly expressed in rodent beta cells during the secondary transition, a period of embryogenesis when beta cell mass expands significantly [112]. Studies from the Newgard lab discovered a role for NKX6.1 in expansion of beta cell mass [113,114]. Upon the overexpression of NKX6.1 in rat islets, they observed a significant increase in the number of cells per islet accompanied by an increase in thymidine and BrdU incorporation, suggesting an increase in cell proliferation. Conversely, they observed that the siRNA knock down of NKX6.1 expression led to a decrease in the incorporation of thymidine. Furthermore, transcriptional analysis showed the upregulation of cell cycle genes, including cyclins A, B and E in response to NKX6.1 overexpression. Importantly, the overexpression of NKX6.1 did not appear to have an adverse effect on glucose stimulated insulin secretion (GSIS), perhaps due to the concomitant upregulation of VGF, which has been shown to enhance GSIS in rat islets [115]. A similar role for NKX6.1 in human beta cells has not yet been demonstrated.

### 6.3. Intracellular Signaling Pathways

A large number of studies in rodent models and immortalized cell lines have characterized the intracellular signaling pathways within the beta cell that could control proliferation in response to mitogenic signals. These studies have been extensively reviewed in a set of three comprehensive perspective articles [116,117,118]. Although these many studies have provided important information regarding the mechanisms through which beta cell proliferation could be induced, many of the findings have not translated to human beta cells, which do not appear to respond to the same signals. Increased accessibility to cadaveric human islets may allow more studies on these pathways on human islets, with the caveat that the ex vivo islet responses may not accurately reflect the in vivo situation.

### 6.4. Small Molecule Induction of Beta Cell Regeneration

In general, the differences in molecular signaling pathways regulating beta cell proliferation in rodents versus humans has made it difficult to translate findings from animal research into clinical therapies. For this reason, efforts are shifting towards high throughput methods of screening small molecules on human islets and or induced pluripotent stem cell (iPSC)-derived islets for potential efficacy. Impressively, three different groups identified several unrelated small molecules—adenosine kinase inhibitor 5-iodotubercidin (5-IT), Harmine and GNF4877—that each promoted beta cell replication through the inhibition of dual-specificity tyrosine-regulated kinase-1a (DYRK1A) [119,120,121]. Subsequent studies of Harmine and its analogs have confirmed that it functions to inhibit DRYK1A and activate calcineurin, effectively promoting beta cell proliferation and the expansion of islet mass [119,120,121,122,123,124]. More recently, the combinatorial inhibition of DRYK1A and Transforming Growth Factor beta (TGFβ) signaling was shown to synergistically promote human beta cell proliferation, without causing adverse functional affects [125]. Although the inhibition of DYRK1A appears to be a potential “magic bullet” for promoting human islet proliferation, challenges remain. One primary issue is that Harmine is known to have psychoactive and hallucinogenic effects on the central nervous system [126]. Furthermore, Harmine and the other small molecules targeting DYRK1A are not beta cell specific and have a number of off target effects. Ultimately, the development of more highly selective beta cell targeting approaches and the generation of more potent DYRK1A-specific antagonists will be necessary for the inhibition of this pathway to be an effective therapy to promote beta cell proliferation.

Another high throughput screening method used a combination of NKX6.1, VGF and proliferative readouts to ensure that GSIS was not impaired upon the induction of replication to identify GNF-9228, a small molecule that stimulated human beta cell proliferation, GSIS and promoted the survival of rat insulinoma cells [127]. Interestingly, GNF-9228 functions to increase beta cell proliferation via a mechanism distinct from DYRK1A inhibitors. Additional screens have utilized adenoviral reporters encoding cell cycle regulators as readouts to identify small molecules that promote human beta cell regeneration. This study identified 12 novel candidate mitogens, including neurotransmitters, hormones, growth factors, and adenosine and TGFβ signaling molecules [122]. With the growing number of high throughput screening tools and a greater appreciation that pathways that could be targeted to promote human beta cell expansion, it is certain that additional pathways and molecular factors will be identified as targets to promote human beta cell proliferation. The new challenge will be determining the specificity and efficacy of these small molecules and drugs in maintaining human beta cell mass in vivo, which will be compounded by the difficulty in measuring beta cell mass in patients over time.

## 7. Exogenous Sources of Beta Cells

### 7.1. Cadaveric Beta Cells

Although the many endogenous approaches addressing decreased beta cell mass in diabetic patients are encouraging, and our knowledge of the mechanisms involved in stabilizing beta cell mass continues to increase, there remains no true cure for diabetes. Currently, the closest approximation to a cure is whole pancreas or pancreatic islet transplantation. Whole pancreas transplantation has been accomplished in several thousand diabetic subjects and has achieved long-term insulin independence and good metabolic control in many individuals [128,129]. Although whole pancreas transplantation has proven to be an effective therapy, the procedure requires major surgery and is associated with a high morbidity rate [130]. Pancreatic islet transplantation has emerged as an alternative treatment with the establishment of the Edmonton protocol by James Shapiro and colleagues [131]. In their initial study, Shapiro et al. performed islet transplantation in combination with glucocorticoid-free immunosuppression on seven T1D patients; all seven patients became insulin independent for at least one year. Prior to this study, the success rate for islet transplantation was only around 8% [132]. The major modification of the Edmonton protocol was the infusion of significantly more islets than previously thought necessary; the mean total beta cell mass per transplant was 132(±67) × 10^6^ islet equivalents (IE) collected from two to three individual donors [131]. They postulated that the transplantation of a larger number of islets was necessary to counteract cell loss due to engraftment failures and/or loss of cells through apoptosis or other nonimmune-mediated inflammation [133,134]. Although this advancement represented a critical treatment break though, it also was associated with new challenges: notably the availability of sufficiently large sources of islets for the transplantation and survival of those cells post transplantation.

The severe shortage of donor islets remains a major hurdle to overcome. The Edmonton protocol requires several donors, and even when donor pancreata are available, the yield of high-quality islets recovered is often very low. The ability to culture ex vivo islets long-term would enable banking and even pooling of low yield islets preps, however, isolated islets tend to rapidly deteriorate under conventional culture conditions. This is likely due to oxygen depletion and consequent necrosis in the core of the islets, and as such, varying the oxygen concentration so that it more closely resembles the in vivo environment could be a key to achieving long-term culture [135,136,137]. While the optimization of islet culture conditions remains a topic of intense research, efforts to maintain islets in vitro for any length of time while maintaining their functionality have thus far failed.

With the challenges associated with procuring sufficient donor islets for transplantation treatments, beta cells derived from human pluripotent stem cells have emerged as a viable and promising alternative. In the past ten years, the field has achieved remarkable advancements in their ability to generate functional beta-like cells for cell replacement therapy, although—as discussed below—several challenges remain.

### 7.2. Mesenchymal Stem Cells

Mesenchymal stem cells (MSCs) are a somatic cell population that is present in many different perinatal and adult tissues (reviewed in [136,137]. Human MSCs represent an attractive stem cell source for the production of pancreatic beta cells due to the fact that they can be isolated from several different tissue sources, are able to self-renew, are relatively easy to culture in vitro, and have the ability to differentiate into several different cell types. The most common source of MSCs is the bone marrow; however, MSCs derived from many different tissues, including adipose tissue, pancreas, Wharton’s jelly of the umbilical cord, dental pulp and placenta have been used with mixed success to produce islet-like cells using a variety of in vitro differentiation protocols (reviewed extensively in [138]. Many of these studies have applied step-wise directed differentiation protocols that mimic normal pancreatic developmental processes—similar to those used to differentiate human pluripotent stem cells (hPSCs) (see below). However, there are a large number of studies suggesting that different cocktails of growth factors can promote the transdifferentiation of MSCs directly into beta-like cells [138]. While several of these studies have produced beta-like cells that can be transplanted into mice to rescue STZ-induced diabetes, the long-term stability of the beta cell phenotype and/or survival of the cells was not explored. A small number of studies have also suggested that undifferentiated bone marrow MSCs that were transplanted into mice could spontaneously migrate to the islet and transdifferentiate into beta cells in vivo [139,140]; however, the conclusions of these studies have been challenged by stronger evidence to suggest that the transplanted MSCs promote islet cell survival, regeneration and function [141,142,143,144,145]. Despite these controversies, MSCs represent a viable alternative source of beta cells. In addition, they have perhaps a more valuable role in facilitating islet survival and growth upon transplantation. The inherent angiogenic and immunosuppressive properties of MSCs have been shown to stimulate and support tissue regeneration and the co-transplantation of MSCs with human islets have improved islet survival [146].

### 7.3. Human Pluripotent Cells: hESCs and iPSCs

Human embryonic stem cells (hESCs) are derived from the inner cell mass (ICM) of the embryo at the blastocyst stage of development [147]. They are characterized by their ability to undergo self-renewal and to differentiate into any cell type in the body; in short, they represent an unlimited source of cells for cell-based therapy. Induced pluripotent stem cells (iPSCs) were first generated in 2007 by the Yamanaka lab in Japan [148]. These iPSCs represent pluripotent cells with similar properties to hESCs, but unlike hESCs they are derived from terminally differentiated cells. Initially, iPSCs were generated from donor skin biopsies and reprogrammed using four transcription factors—OCT4, cMYC, KLF4 and SOX2 (known as the Yamanaka factors)—to become bone fide stem cells. These cells are pluripotent and proliferative, and most significantly, retain the genetic profile of the person from which they were derived. These features make it possible to take adult somatic cells from any patient, reprogram them to iPSCs, and then differentiate them to a cell type of interest, such as islet beta cells, that can then be transplanted without immune consequences or the need for immunosuppression.

To truly harness the potential of human pluripotent stem cells (hPSCs) (either hESCs or iPSCs) as a therapy for diabetes, there is a need to develop reproducible and efficient protocols that consistently promote their directed differentiation into the different islet cell types, and primarily beta cells. Despite significant progress, even the most advanced differentiation protocols fail to produce a full complement of mature islet cells that are phenotypically and functionally similar to those found in healthy human islets. Additionally, these protocols suffer from inconsistent reproducibility between differentiations and, similar to cadaveric islets, there are also issues with cell survival upon transplantation.

The most successful protocols that have been developed for differentiating beta cells from hPSCs faithfully recapitulate the developmental cues that a cell would normally receive in an in vivo context (Figure 2). As our knowledge of human pancreas development is limited, directed differentiation of hPSCs to stem cell-derived beta cells (sBCs) is largely based on knowledge derived from animal models of pancreas development and disease. Initial protocols from several labs succeeded in producing cells that phenotypically resembled primary human islets and were capable of reversing diabetes in rodent models [149,150,151]; however, none of these protocols resulted in a uniform population of fully functional beta cells [152]. In each case, these sBCs were found to be immature, resembling human fetal beta cells as defined by their gene expression profiles, truncated dynamic glucose stimulated insulin secretion (dGSIS) profiles (reviewed in Figure 3), insulin processing efficiency (as demonstrated by an elevated proinsulin/insulin ratio) and lower total insulin content [150]. However, upon long-term transplantation in diabetic mice, the immature sBCs appeared to gain functionality and were able to rescue the diabetic phenotype [149,150,151].

More recently, however, additional studies from Velazco-Cruz and Nair developed protocols that enhanced the functionality of sBCs in vitro by altering culture conditions and/or recapitulating endocrine cell clustering. In both studies, these protocol modifications resulted in the production of more phenotypically mature sBCs in vitro with dynamic glucose stimulated insulin secretion (dGSIS) profiles that were comparable to human islets [153,154]. Interestingly, although these sBCs were demonstrated to have more mature functionality in a number of ways, the sBCs produced in either study did not express *MAFA* or *UCN3*—two genes that are normally associated with adult functionally mature beta cells.

One ongoing challenge in the generation of mature sBCs is the identification of specific markers that indicate the production of mature human beta cells. As mentioned above, one such marker, UCN3, is postulated to be a key marker of mature of human beta cells. In mice, UCN3 is highly and specifically expressed in adult beta cells where it is required for glucose stimulated insulin secretion [155,156]. Detailed immunohistochemical analysis has revealed that UCN3 protein is first detectable at embryonic day (E) 17.5 and from postnatal day seven is an exclusive marker of mouse beta cells [157]. Similar analysis combined with gene expression experiments showed that UCN3 is expressed in adult human and non-human primate beta cells though not exclusively, as it also appears in alpha cells [157]. Van der Meulen et al. also showed that hESC-derived pancreatic endoderm organoids engrafted into mice express UCN3 on maturation to endocrine cells [157]. This study, while confirming UCN3 as a marker of mature endocrine cells, also highlighted the variability in marker expression between species. UCN3 expression in sBCs is also somewhat variable and does not always correlate with apparent functional maturity [152]. However, a recently developed differentiation protocol using a high content chemical screen has succeeded in producing beta-like cells that do express UCN3 [158]. Similar to other differentiation protocols, these cells demonstrated the effective rescue of blood glucose levels on transplantation into diabetic mice, but they failed to show a mature dGSIS response in vitro, suggesting that despite a mature phenotype at the gene expression level, they may not have actually achieved functional maturity. The identification of a truly specific (and universal) biomarker for mature human beta cells would be an invaluable tool for assessing sBC maturity and potential functionality. In efforts to achieve this goal, a recent study by Augsornworawat et al. performed single cell transcriptome profiling of sBCs post-transplantation [159]. This study validated many of the known markers associated with beta cell maturity and provides a resource that can be mined for novel beta maturation markers.

The advent of single cell RNA sequencing (scRNAseq) technology in recent years has also revealed the extensive heterogeneity of sBC cultures and has shed light on the differentiation processes that give rise to them. In 2019, Veres et al. published transcriptional profiling of over 100,000 human cells at various stages of in vitro beta cell differentiation [160]. Their extensive analysis of the developmental order in which cell types arise in vitro and the dynamic molecular alterations that drive each differentiation trajectory create an in-depth road map for beta cell differentiation that will inform future differentiation protocols. Furthermore, they identify a novel surface marker for sBCs, ITGA1, that they postulate could be used to isolate cells for use in cell therapy and drug testing [160]. scRNAseq analysis of sBCs before and after in vivo maturation in mice has also provided insights into the significant transcriptional changes that occur during the maturation process and are associated with improved functionality [159].

Although, as described above, there has been tremendous progress with the in vitro generation of human sBCs, protecting these cells upon transplantation will be the next hurdle to overcome. Similar to transplanted cadaveric islets, sBCs undergo extensive cell death upon transplantation. A study by Feleo et al. demonstrated that culturing sBCs in physiological oxygen concentrations and supplementing the transplanted cells with amino acids prevented hypoxia-induced death and improved cell survival [161]. Based on the normal conditions that islets usually thrive in, it has also been speculated that co-transplantation of the sBCs with endothelial cells, mesenchymal stem cells, neuronal cells and/or components of the extracellular matrix will also promote their survival and function, although studies so far have met with mixed results, this remains an understudied area of research and there have yet to be extensive direct comparisons of the efficacy of accessory tissues and/or cells [162,163,164].

In addition to developing protocols to improve sBC graft survival post-transplantation, it is also necessary to protect transplanted cells from immune rejection. With the advent of using individualized patient-derived iPSCs as the source for beta cells, this has become less of an issue; however, this remains a costly and time-consuming effort. Furthermore, for individuals with T1D, the transplanted sBCs also need to be protected from recurrent autoimmune destruction. Historically, immunosuppression has been the predominant therapy used to protect the graft, although it has its associated risks [165]. More recently, several groups have explored the use of encapsulation devices to physically protect the sBCs from the immune system. Several different biomaterials have been developed that also incorporate chemical modifications to increase biocompatibility and have the ability to release factors to favorably support graft survival (reviewed in [166,167]). In recent years, there have been extraordinary advancements in the development of several different islet encapsulation technologies; however, each modality has unique limitations that currently impede their extensive use in patients [166]. However, the rapid pace at which the biomaterials field is advancing suggests that cell encapsulation therapies will be a viable treatment option in the near future.

Finally, an exciting area of emerging research is the use of gene editing technologies to engineer immune privileged beta cells. This approach has been used successfully to overexpress the immune checkpoint protein programmed death ligand 1 (PD-L1) in iPSC-derived human islet-like organoids to protect them from xenograft and allogenic rejection [168]. Additional approaches currently under investigation include the deletion of MHC class I and II genes to reduce the alloimmune response against beta cells or by modulating the pathways involved in the beta cell inherent inflammatory response. More recently, Cai, et al. used an unbiased CRISPR/CAS9 screen to identify Renalase (RNLS) as a novel target for making beta cells resistant to autoimmune destruction [169]. These studies are still in their formative stages, but with the success of immune-modulatory technologies in cancer treatments, the development of immune-protected sBCs could move them a step closer to being a viable regenerative therapy for diabetes.

## 8. Conclusions

In summary, major inroads have been made towards the development of regenerative therapies for the treatment of diabetes (Figure 4). As with most fields, progress is constantly impeded by an incomplete understanding of the inherent biology of the pancreatic beta cells; however, the more knowledge we gain, the greater chance we have of overcoming any barriers. This is nicely exemplified by the discovery of the remarkable plasticity of mature endocrine cells and the ongoing development of novel approaches that promote the limited regeneration of endogenous beta cells. Furthermore, this knowledge has led to the rapid development of improved protocols to generate exogenous human beta cells from stem cell sources and the derivation of novel approaches to promote the survival and immune protection of these cells upon transplantation. While several hurdles still remain, these major advances suggest that beta cell regeneration will present itself as a viable cure for diabetes in the not-too-distant future.

## Figures and Tables

**Figure 1 ijms-22-03306-f001:**
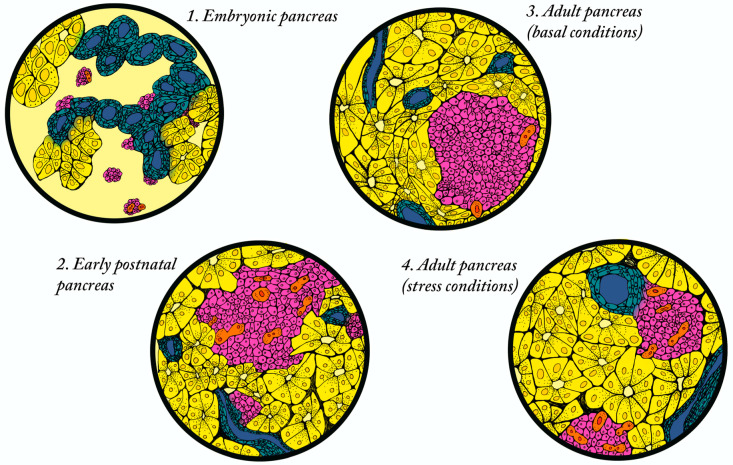
Different mechanisms of endogenous beta cell regeneration exist throughout life. (**1**) Embryonic pancreas. During embryonic development, the beta cells primarily arise from the Neurogenin3 (NEUROG3)-expressing endocrine progenitor population located in epithelial ducts through a mechanism of neogenesis. (**2**) Early postnatal pancreas. In the first two weeks after birth, the beta cell population expands through a combination of neogenesis and replication of existing beta cells. (**3**) Adult pancreas in basal conditions. In the adult, a limited number of new beta cells form through replication. (**4**) Adult islet in stressed conditions. Adult beta cells exposed to physiological stresses (pregnancy, obesity, insulin resistance, damage) regenerate through both neogenesis (rare) and the replication of existing beta cells. Key: beta cells (pink); epithelial ducts (blue); replicating beta cells (orange); acinar cells (yellow).

**Figure 2 ijms-22-03306-f002:**
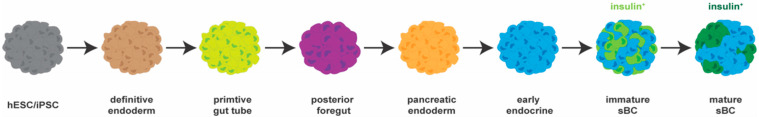
Stem cell-derived beta-like cell differentiation protocols recapitulate developmental cues. Human pluripotent stem cells (hPSCs) are characterized by the expression of OCT4, NANOG and SOX2. The most successful beta cell differentiation programs recapitulate normal development. The first step includes a transition to a definitive endoderm (DE) cell fate that is driven by activation of the WNT and TGFβ pathways. After two days of exposure to the small molecules CHIR (WNT agonist) and Activin-A (TGFβ agonist), the cells express transcription factors specific to the definitive endoderm (DE) stage of development including SOX17. Next, the DE cells are differentiated towards the primitive gut tube (PGT) by the activation of the fibroblast growth factor (FGF) signaling pathway with KGF, a process that takes approximately three days. For the cells to activate the key pancreatic transcription factor, PDX1, and become specified to the posterior foregut (PF) they must receive cues that activate retinoic acid and hedgehog signaling and cues that inhibit Bone Morphogenetic Protein (BMP) signaling. At this stage, the cells are primed to be differentiated into any cell type in the pancreas and will become more fate restricted with time. The PDX1 positive PF cells are then differentiated into pancreatic endoderm (PE) cells that express PDX1 and NKX6.1 by activation of the FGF signaling pathway with EGF and KGF. These cells then undergo endocrine differentiation, during which they are exposed to signals that direct them to pancreatic beta cell fate while blocking differentiation to other pancreatic cell types. The key factors involved at this stage block TGFβ, BMP and Notch signaling. As they progress from the PE stage, the cells transiently express transcription factor NGN3 and then become NKX2.2 expressing early endocrine cells (EEs). NKX2.2 induction has been shown in 90–95% of cells at this stage demonstrating that early endocrine differentiation can be performed at very high efficiency [151]. After ~ ten days of endocrine differentiation, the cells express the key beta cell markers and begin to produce the insulin hormone, however, they are not yet considered functional as they do not regulate insulin secretion in response to glucose stimulation. A further five–seven days of maturation in a minimal media is required to allow the cells to become functional immature stem cell-derived beta-like cells (sBCs). Additional maturation steps, such as in vitro aggregation or transplantation into mice or humans, is required to achieve functionally mature sBCs. iPSC: induced pluripotent stem cell; hESC: human embryonic stem cells.

**Figure 3 ijms-22-03306-f003:**
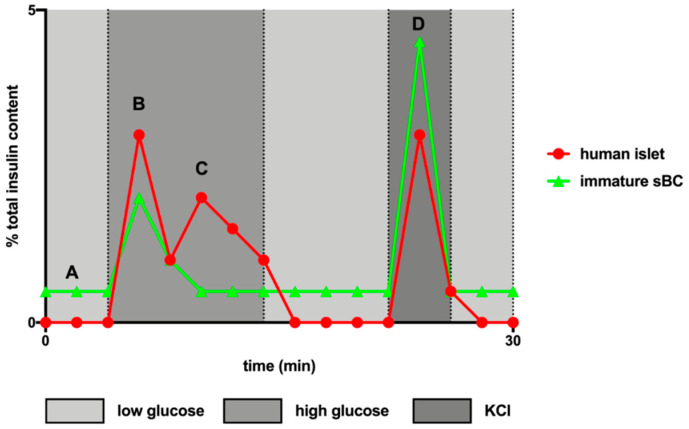
Dynamic glucose stimulated insulin secretion profiles depend on the maturation state of beta cells. Schematic representation of idealized dynamic glucose stimulated insulin secretion (dGSIS) profiles for cadaveric human islets 24 h post retrieval and immature sBCs obtained by perifusion. The values are presented as a percentage of total insulin content of the cluster of cells recovered following perifusion. Low glucose is 0.5 mM glucose in Krebs-Ringer Bicarbonate (KRB) buffer, high glucose is 16.7 mM glucose in KRB buffer and KCl is 30 mM KCl + 16.7 mM glucose in KRB buffer. A, Immature sBCs present with an increased glucose secretion in basal glucose. B, Immature sBCs respond to high glucose though often not to the same degree as human islets. C, Immature sBCs lack the characteristic second phase response to high glucose observed in human islets. D, Immature sBCs often show an exaggerated or so-called “uncontrolled” response to membrane depolarization with KCl; while human islets respond by secreting insulin at a level similar to their maximum response to high glucose.

**Figure 4 ijms-22-03306-f004:**
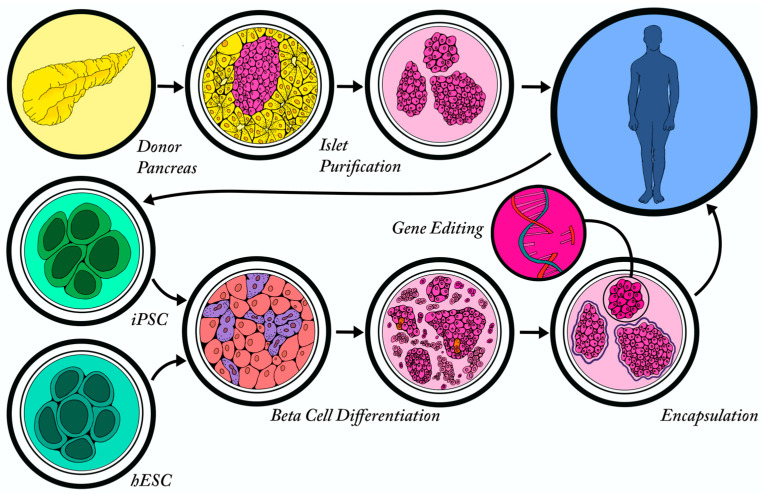
Exogenous sources of beta cells. Exogenous beta cells can be purified from cadaveric beta cells isolated from a donor pancreas and transplanted into a T1D patient following the Edmonton protocol. Alternatively, terminally differentiated cells can be isolated from an individual and reprogrammed into iPSCs, followed by directed differentiation in vitro to form stem cell-derived beta-like cells (sBCs). In addition, hESCs can be differentiated to form non-patient specific sBCs. The sBCs from both iPSCs and hESCs can be genetically manipulated or encapsulated to help them evade the patient’s immune system following transplantation.

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
