# Peer review of "Islet Regeneration: Endogenous and Exogenous Approaches"

_ijms, 2021, doi:10.3390/ijms22073306_

Round 1

Reviewer 1 Report

This is a well written article. However, discussion on islet de-/regeneration  without changing islet mass induced by defective breathing and metabolic disorders were omitted.  Paragraphs elaborating islet dysfunctions induced by environmental factors and metabolic defects should be included. 

Author Response

We have only addressed the concern about how metabolic defects (obesity and insulin resistance) influence beta cell mass. I am not sure how breathing disorders fits into this review. In addition, the review is focused on regenerative approaches rather than focusing on islet dysfunctions so we did not add additional information on this topic. 

Reviewer 2 Report

In this review, Docherty and Sussel summarize current approaches to increase beta-cell mass as a potential therapy for diabetes. This is a well-written and comprehensive review that will contribute to the ongoing discussion on this subject.

I have few minor comments:

1) Figure 1 does not reflect the differences in beta-cell replication rate between the different conditions - it appears as more beta-cells undergo division in the adult islets as compared to the neonatal and stress islets. Also, neonatal islets tend to be not as round as adult ones. Please correct this figure.

2) There are studies from several groups on the extrinsic regulation on beta-cell replication during neonatal and stress conditions by both systemic factors (f.i., SerpinB1 ) and cells of the islet niche (macrophages, endothelial cells, pericytes). Please include a paragraph that addresses this layer of regulation.

3) Throughout the text: please adhere to the proper nomenclature of rodent vs. human proteins and genes (f.i., mouse Neurog3 and not NEUROG3, line 82)

Author Response

We appreciate the reviewers comments and have responded below.

1) Figure 1 does not reflect the differences in beta-cell replication rate between the different conditions - it appears as more beta-cells undergo division in the adult islets as compared to the neonatal and stress islets. Also, neonatal islets tend to be not as round as adult ones. Please correct this figure.

The figure has been corrected accordingly.

2) There are studies from several groups on the extrinsic regulation on beta-cell replication during neonatal and stress conditions by both systemic factors (f.i., SerpinB1 ) and cells of the islet niche (macrophages, endothelial cells, pericytes). Please include a paragraph that addresses this layer of regulation.

We have added a new section (3 paragraphs) on the islet niche signals. And included additional information on systemic factors to the metabolic stress (obesity) to discuss systemic factors that influence proliferation.

3) Throughout the text: please adhere to the proper nomenclature of rodent vs. human proteins and genes (f.i., mouse Neurog3 and not NEUROG3, line 82)

We have gone through and corrected where necessary. In mice proteins are upper case, similar to humans so Line 82 is referring to the mouse NEUROG3 protein. When referring to gene expression we used Neurog3.

Round 2

Reviewer 1 Report

As an alternative beta cell source, mesenchymal stem cell derived beta cells should be addressed.

  1. Apart from beta cell mass proliferation, beta cell function and dysfunction due to respiratory (mitochondrial influence) stress under hypoxia should be discussed. This issue has to do with regeneration after islet transplantation.
  2. As an alternative beta cell source, mesenchymal stem cell derived beta-cells should be addressed. Review Govindasamy V et al. (2010), Kim G et al. (2016) and the like.

Author Response

We have added a paragraph describing the use of MSCs as an exogenous stem cell source for islets.  We have chosen not to include information related to regeneration of islets upon transplantation since we believe this is beyond the scope of this review and most of the studies related to this issue suggest that prevention of hypoxia post-transplantation supports beta cell survival rather than promoting proliferation.